# Profiling Colorectal Cancer in the Landscape Personalized Testing—Advantages of Liquid Biopsy

**DOI:** 10.3390/ijms22094327

**Published:** 2021-04-21

**Authors:** Donatella Verbanac, Andrea Čeri, Iva Hlapčić, Mehdi Shakibaei, Aranka Brockmueller, Božo Krušlin, Neven Ljubičić, Neven Baršić, Dijana Detel, Lara Batičić, Lada Rumora, Anita Somborac-Bačura, Mario Štefanović, Ivana Ćelap, Alma Demirović, Roberta Petlevski, József Petrik, Marija Grdić Rajković, Andrea Hulina-Tomašković, Ivana Rako, Luciano Saso, Karmela Barišić

**Affiliations:** 1Department of Medical Biochemistry and Hematology, Faculty of Pharmacy and Biochemistry, University of Zagreb, Ante Kovačića 1, 10000 Zagreb, Croatia; andrea.ceri@pharma.unizg.hr (A.Č.); iva.hlapcic@pharma.unizg.hr (I.H.); lada.rumora@pharma.unizg.hr (L.R.); asomborac@pharma.unizg.hr (A.S.-B.); mario.stefanovic@kbcsm.hr (M.Š.); roberta.petlevski@pharma.unizg.hr (R.P.); jozsef.petrik@pharma.unizg.hr (J.P.); mgrdic@pharma.unizg.hr (M.G.R.); andrea.hulina@pharma.unizg.hr (A.H.-T.); karmela.barisic@pharma.unizg.hr (K.B.); 2Musculoskeletal Research Group and Tumour Biology, Faculty of Medicine, Institute of Anatomy, Ludwig-Maximilian-University Munich, Pettenkoferstrasse 11, D-80336 Munich, Germany; mehdi.shakibaei@med.uni-muenchen.de (M.S.); Aranka.Brockmueller@med.uni-muenchen.de (A.B.); 3School of Medicine, University of Zagreb, Šalata 3, 10000 Zagreb, Croatia; bozo.kruslin@mef.hr (B.K.); neven.ljubicic@kbcsm.hr (N.L.); neven.barsic@gmail.com (N.B.); 4Department of Pathology and Cytology “Ljudevit Jurak”, University Hospital Centre “Sestre milosrdnice”, University of Zagreb, Vinogradska 29, 10000 Zagreb, Croatia; alma.demirovic@kbcsm.hr; 5Department of Internal Medicine, University Hospital Centre “Sestre milosrdnice”, Division of Gastroenterology and Hepatology, University of Zagreb, Vinogradska 29, 10000 Zagreb, Croatia; 6School of Dental Medicine, Gundulićeva 5, 10000 Zagreb, Croatia; 7Department of Medical Chemistry, Biochemistry and Clinical Chemistry, Faculty of Medicine, Braće Branchetta 20/1, 51000 Rijeka, Croatia; dijana.detel@medri.uniri.hr (D.D.); lara.baticic@medri.uniri.hr (L.B.); 8Department of Clinical Chemistry, University Hospital Centre “Sestre milosrdnice”, University of Zagreb, Vinogradska 29, 10000 Zagreb, Croatia; ivana.celap@gmail.com; 9Department of Laboratory Diagnostics, University Hospital Centre Zagreb, University of Zagreb, Kišpatićeva 12, 10000 Zagreb, Croatia; ivana.rako@kbc-zagreb.hr; 10Department of Physiology and Pharmacology “Vittorio Erspamer”, Sapienza University of Rome, Piazzale Aldo Moro 5, 00185 Roma, Italy; luciano.saso@uniroma1.it

**Keywords:** biomarkers, colorectal cancer, early detection examination, liquid biopsy, personalized medicine, tumor treatment, exosomes, ctDNA, CTC

## Abstract

Drug-specific therapeutic approaches for colorectal cancer (CRC) have contributed to significant improvements in patient health. Nevertheless, there is still a great need to improve the personalization of treatments based on genetic and epigenetic tumor profiles to maximize the quality and efficacy while limiting cytotoxicity. Currently, CEA and CA 19-9 are the only validated blood biomarkers in clinical practice. For this reason, laboratories are trying to identify new specific prognostics and, more importantly, predictive biomarkers for CRC patient profiling. Thus, the unique landscape of personalized biomarker data should have a clinical impact on CRC treatment strategies and molecular genetic screening tests should become the standard method for diagnosing CRC. This review concentrates on recent molecular testing in CRC and discusses the potential modifications in CRC assay methodology with the upcoming clinical application of novel genomic approaches. While mechanisms for analyzing circulating tumor DNA have been proven too inaccurate, detecting and analyzing circulating tumor cells and protein analysis of exosomes represent more promising options. Blood liquid biopsy offers good prospects for the future if the results align with pathologists’ tissue analyses. Overall, early detection, accurate diagnosis and treatment monitoring for CRC with specific markers and targeted molecular testing may benefit many patients.

## 1. Introduction

Colorectal cancer (CRC) represents the second most common cause of cancer-related death globally [1], with annual incidence approaching two million cases worldwide [2] (Figure 1). Moreover, CRC incidence is rising in low-income and middle-income countries [1]. The disease results from the accumulation of multiple genetic and epigenetic modifications that lead to the transformation of colonic epithelial cells into invasive and aggressive adenocarcinomas [3,4]. The lack of and inadequate response to numerous mono-target therapies in cancer treatments emphasizes that personalized diagnostic and therapeutic approaches are necessary for effective strategies that target not only tumor cells, but more importantly, the multicellular tumor microenvironment for improved patient outcomes. Nevertheless, one of the most important keys to successful treatment of this malignant tumor and patient survival is not only the early diagnosis of the disease but also controlling tumor dissemination and progression [5]. For example, the 5-year survival rate for patients with early diagnosis is approximately 90%. In contrast, the survival rate for patients with regional lymph node metastasis is around 70%, and for those with distant metastases it is only 13% [6,7].

Other important keys to improving CRC therapies are enhancements in surgical modalities and adjuvant chemotherapy, which has increased the cure rates in early-stage disease. Still, unfortunately, a significant proportion of patients will develop recurrence or advanced illness. Nevertheless, the efficacy of chemotherapy for recurrence and advanced CRC has improved significantly over the last decade. Previously, the historical drug 5-fluorouracil was the only chemotherapeutic agent used. With the addition of other chemotherapeutic agents such as capecitabine, irinotecan, oxaliplatin, bevacizumab, cetuximab, panitumumab, vemurafenib, and dabrafenib, the median survival of patients with oligometastatic CRC has improved significantly from less than one year to the current standard of nearly two years [8]. However, many side effects of systemic therapy, such as toxicity, may cause fatal complications and significantly affect the patients’ quality of life. In parallel, a plethora of biologically active compounds are tested in vitro and in vivo and promising hits/leads compounds that may be used in the development as adjunct to the therapy are continuously identified [9,10]. An overview of existing CRC-targeted agents and their underlying mechanisms, as well as a discussion of their limitations and future trends, has been published recently [11]. Still there is an urgent need for crucial biomarkers to select optimal drugs individually or in combination for an individual patient. The application of personalized therapy based on DNA testing could help clinicians provide the most effective chemotherapy agents and dose modifications for each patient. Yet, some of the current findings are controversial, and the evidence is conflicting [12]. The current trend is to achieve successful personalized therapeutic approaches based on monitoring of disease-specific biomarker(s). However, the data in this respect is scarce and studies which include the personalized testing vs treatment are needed. The aim of our ongoing translational research is to contribute to this unmet medical need.

## 2. Etiology of Colorectal Cancer

The etiology of CRC is extensively described in the literature but is still not known in detail [13]. In this review, we describe specifically the CRC-related genes and pathways, while knowing that they often overlap with other solid tumors, such breast and prostate cancer. In approximately 70–90% of patients, CRC develops sporadically due to point mutations of the *APC*, *KRAS*, *TP53*, and *DCC* genes.

In approximately 1–5% of cases, it is a consequence of a hereditary polypoid and non-polypoid syndrome and 10–30% of patients have a familial CRC [14] (Figure 1). It is important to note that 1–2% of CRC has been associated with chronic inflammatory conditions such as ulcerative colitis and Crohn’s disease. The risk increases with the longer duration of ongoing inflammation [15] caused also by the dysbiosis in the gut [16] and inappropriate nutrition patterns and deteriorating life-style conditions [17]. Chromosomal instability (CIN), as an essential molecular pathway of malignant transformation, mainly affects genes such as *APC*, *KRAS*, *PIK3CA*, and *TP53* [18]. In addition, the adenoma–carcinoma sequence offers potential for screening and surveillance; e.g., connexin 43 expression in colonic adenomas is linked with high-grade dysplasia and colonic mucosa surrounding adenomas [19]. *APC* mutations lead to nuclear beta-catenin translocations and the transcription of genes participating in carcinogenesis and invasion processes. *KRAS* and *PIK3CA* mutations lead to continuous activation of mitogen-activated protein kinase (MAPK) pathways, which, in turn, increases cell proliferation, whereas *TP53* mutations lead to the loss of the p53 function and uncontrolled cell cycle [18]. Finally, epigenetic instability (CIMP) is associated with hypermethylation of the oncogenes promotor and loss of expression of the corresponding proteins [20].

## 3. The impact of Genetic Alterations on Disease Outcome

The most common mutations, chromosomal alterations, and translocations affect critical wingless-related integration sites (WNT), MAPK/phosphatidylinositol 3-kinase (PI3K), and transforming growth factor β (TGF-β) signaling pathways and intracellular protein functions such as p53, as well as cell cycle regulation [21]. The WNT pathway, which is a critical mediator of tissue homeostasis and repair, is frequently co-opted during tumor development. Almost all colorectal cancers demonstrate hyper-activation of the WNT pathway, which is considered to be the initiating and driving event in many cases [22]. *APC* gene mutations represent the most significant genetic change associated with the WNT signaling pathway, regulating stem cell differentiation and cell growth. Nevertheless, they do not represent a good predictor of the disease progression due to their high CRC frequency and the number of various mutations identified within the gene [23]. Increased β-catenin expression associated with the WNT signaling pathway has also been recognized as a non-reliable marker for disease prognosis. In contrast, overexpression of the *c-MYC* gene triggered by the activation of WNT signaling pathway represents a good predictor of metastasis and disease progression [24,25]. *KRAS*, *BRAF*, and *PIK3CA* mutations are common and are associated with the MAPK/PI3K signaling pathways.

Furthermore, mutations of the *KRAS* gene in exon 2, codon 13 are associated with poor prognosis and a low survival rate, while mutations in exon 2 and codon 12 are associated with tumor progression and metastasis [26,27]. Recently, AMG 510, the first *KRAS* G12C inhibitor, after promising preclinical results, has entered into the clinical development [28]. This represent fascinating efforts that could overcome the perception that *KRAS* is in principle "undruggable" as a therapeutic target and may contribute to the development of effective drugs for targeting traditionally difficult signaling pathways in the clinical setting [29]. *BRAF* gene mutations are associated with poor prognosis and survival [30,31,32].

The associations between disease outcome or survival and *PIK3CA* mutations have not yet been established. Still, evidence supports that these mutations, combined with the *KRAS* gene mutations, are associated with poor outcomes [33]. Furthermore, CRC patients with multiple *PIK3CA* mutations, e.g., a combination of mutations in exons 9 and 20, have a poorer prognosis than patients with only one of these mutations [34]. Protein phosphatase and tensin homolog (PTEN) adversely affects the PI3K signaling pathway and CRC in which the loss of the *PTEN* gene has been associated with a poor prognosis [35]. In CRC patients, changes in the TGF-β signaling pathway are associated with CIN [36]. Chromosome 18q is bearing the tumor suppressor genes *SMAD2* and *SMAD4* and their encoded proteins are functionally associated with apoptosis and cell cycle regulation [37,38]. Likewise, they play a role in tumor cell migration by regulating the activity of proteins such as matrix metallopeptidase 9 (MMP9) [39]. A significant association between the loss of chromosome 18q and poor prognosis and survival has not been found [37,38]. In CRC, the loss of the 17q-*TP53* gene, which encodes a tumor suppressor protein p53 that regulates the cell cycle, is quite common. Without it, cells proliferate uncontrollably and tumor progresses [40]. Janus kinases, JAK1 and JAK2, are associated with cytokine receptors [41,42], and cytokine binding leads to their activation and phosphorylation. Afterwards, Janus kinases phosphorylate signal transducer and activator of transcription (STAT) proteins, leading to their translocation to the nucleus and transcription of their target genes [41,42]. There is evidence of JAK1 and JAK2 gene mutations that inhibit the function of the corresponding JAK1 and JAK2 proteins. The JAK1 frameshift mutations (positions 142/143, 430/431, and 860/861) have been described as the consequence of insertion/deletion of one nucleotide [42]. The V617E mutation leads to the JAK2 loss-of-function mutation [41]. These mutations have been found in tumors with high MSI resulting from dysfunctional DNA repair during replication, known as mismatched repair [43]. Indeed, they were associated with tumor resistance to treatment targeting the programmed cell death protein 1 (PD-1) [43]. Determination of molecular changes at the DNA level, particularly derived from tumor-specific liquid components such as circulating tumor DNA (ctDNA), exosomes and circulating tumor cells (CTC), can improve prediction of disease development and help in adjustment of therapy for each patient individually, as part of personalized health care.

## 4. Liquid Biopsy

Much has been learned about the molecular background of CRC development and progression, which may help to tailor therapy for each patient and improve their survival prognosis. However, advances in early CRC diagnosis have not been made as far as expected, and still rely on biomarkers from readily available biological materials (Table 1). Currently, there are only two validated protein-based blood biomarkers used in routine clinical practice: carcinoembryonic antigen (CEA) and carbohydrate antigen 19-9 (CA 19-9). CEA is an embryo-specific glycoprotein that can also be found in CRC. In clinical practice, it is used to monitor the tumor’s progression after its diagnosis [44]. However, it shows an insufficient sensitivity and specificity since it is hereditarily determined and in the case of recessive homozygote, the levels of CA 19-9 would not be increased (approximately in 15% of individuals) [45,46,47]. During the search for new biomarkers that would replace the old ones, a promising non-invasive, and a repeatable procedure called “liquid biopsy” was developed for different body fluids (blood, saliva and urine). Peripheral blood liquid biopsy is used for diagnostic screening, as well as for determining a response to therapy and evaluating the outcome of the disease [48]. Peripheral blood can contain CTC, ctDNA and exosomes (vesicular structures, which contain proteins and RNA molecules, that may be released into circulation by different cells, including tumor cells) [49]. This could make it possible to determine the molecular profile of the disease, the degree of affected tissue, and the response to therapy in a non-invasive way. The founder and establisher of new principles and methods of healing, Leroy Hood, has relentlessly emphasized that in the new era of personalized approaches undertaken while assessing different conditions and diseases, “the blood becomes a window through which we observe what is happening in the body” [50]. The same idea was accepted and maintained by other biomedical disciplines, from genetics to personalized nutrition [51]. Future molecular profiling, ideally assessed and monitored by liquid biopsy, might personalize decision-making even more in CRC patients’ adjuvant scenery [52,53]. 

However, liquid biopsy results need to be combined and evaluated with the tissue’s pathological findings before final validation of the proposed approach. The existing testing landscape presents additional challenges in the application of liquid biopsy in clinical practice, and consideration needs to be given to how the pathologist should be involved in interpreting liquid biopsy data in the context of the patient’s cancer diagnosis and stage assessment [54].

## 5. Circulating Tumor DNA (ctDNA)

Tumor cells release small DNA fragments through various mechanisms, including apoptosis, necrosis and active secretion from tumor cells [60]. These single- or double-stranded DNA fragments in the circulation may contain cancer-related gene mutations such as point mutations, copy number variations, chromosomal rearrangements, and DNA methylation. ctDNA reflects the genetic and epigenetic properties of the genomic DNA in tumor cells [61]. Therefore, identical variants present in the genomic DNA of tumor cells, such as mutations in *KRAS*, *NRAS*, *BRAF*, and other genes, can be identified in ctDNA. Furthermore, due to a relatively short half-life and therefore a rapid turnover in the circulation, ctDNA is considered a real-time biomarker of mutation dynamics and tumor burden [57]. The concentration of ctDNA is not related to a specific type of tumor, its size, or stage of progression, although there are reports of higher ctDNA levels in patients with advanced disease and distant metastases [58]. ctDNA can represent between 0.01% and 90% of total cell-free circulating DNA (cfDNA) [61,62]. The concordance between cfDNA within liquid biopsy and genomic DNA within tumor tissue biopsy is still under debate. Kang et al. compared somatic mutations of the 10 genes between cfDNA and genomic DNA from CRC metastatic tumor tissues and observed an overall 93% concordance rate between the two types of samples [63].

On the other hand, there is evidence that some types of tumor, like gliomas or sarcomas, are not good shedders of ctDNA, although the reason is still unclear [64]. Numerous methods for the analysis of free, non-cellular, circulating DNA in the diagnosis of tumors have been used (Figure 2) [65,66]. However, in clinical practice, their value has not been fully known and accepted yet. Additional studies and data are needed to evaluate the potential and significance of ctDNA further and move it into the clinical mainstream. Currently, several clinical trials are ongoing [67]. Our goal is to point out the importance and provide additional justification to include such procedures in clinical analyses. We hope that in the near future, a faster, non-invasive, more timely, less costly diagnosis of CRC and other malignant tumors will be possible.

## 6. Exosomes

Exosomes, together with apoptotic bodies and microvesicles, belong to the group of extracellular vesicles. They are considered as nanovesicles (because of their diameter between 30 and 120 nm), composed of a phospholipid bilayer, originating from multivesicular bodies generated during the endocytic cycle [68,69]. There is evidence of their potential role in various biological events, such as in intercellular communication [70], cell signaling [71], tissue regeneration [72], immune response [73], cancer development [74], and metastasis [75]. They have a unique capability to transfer different contents including DNA, RNA and proteins. Exosomes may contain different heat shock proteins (Hsps) (Hsc70, Hsp70, Hsp60, and Hsp90) [76,77], which mediate protein distribution in intraluminal vesicles (exosome precursors) and inclusion of cytoskeleton proteins such as actin, tubulin and cofilin [78]. Exosomes also express proteins from the dipeptidyl-peptidase IV (DPP IV) and MMP9 families, involved in the extracellular matrix remodeling, representing the reason why exosomes are associated with tumor invasion and metastasis [59]. They are usually enriched with lipid rafts containing cholesterol, sphingolipids, ceramides, and glycerophospholipids with long-chain saturated fatty acids [70].

Evidence shows that exosomes originating from the human colon carcinoma cell line LIM1215 contain A33 antigens and epithelial cell adhesion molecules (EpCAM), also known as cluster of differentiation 326 (CD326), molecules specific for colonic epithelial cells [79]. It is worth mentioning that the A33 antigen is a glycoprotein highly expressed in CRC [80], while EpCAM expression is increased in most CRCs. EpCAM is significantly associated with uncontrolled cell proliferation and CRC invasion, and metastasis [81].

CRC-generated exosomes contain intracellular CRC proteins [82]. Some of them, such as cadherins, CEA, and TGF-β, may be used for early detection of CRC [82,83]. Recent protein analysis of exosomes isolated from the blood of CRC patients and the blood of healthy volunteers showed that the levels of proteins involved in the remodeling of the extracellular matrix, intercellular communication, and cell signaling, increased vascular permeability, and tumor-promoting inflammation (α-1 antitrypsin (SERPINA1), α -2 antiplasmin (SERPINF2), and MMP9), are increased in CRC patients [84]. In contrast, the level of proteins involved in immune evasion, complement binding, cell adhesion, and tumor growth (integrin-linked protein kinase (ILK), calpain small subunit 1 (CAPNS1), and neuroblastoma RAS (NRAS)) were decreased, although many of them are known to show higher expression in tumor tissue [84]. The proteome profile of exosomes generated from human metastatic colon cancer cells SW620 differs significantly from the proteome profile of exosomes found in non-metastatic primary CRC [85]. Exosomes derived from metastatic CRC contained ample amounts of metastatic factors, signaling molecules, lipid rafts and their associated elements [85].

In general, exosomes contain different RNA molecules, including mRNA, microRNA, long non-coding RNA (lncRNA), and circular RNA (circRNA). Elevated levels of exosomal microRNA (namely, miR-17-92a, miR-19a, miR-210, miR193a) were found in invasive metastatic tumors and are associated with poor prognosis [86].

## 7. Circulating Tumor Cells (CTC)

Circulating tumor cells (CTC) are a very rare subset of cells found in the blood of patients with solid tumors. One milliliter of tumor patients’ peripheral blood may contain approximately ten CTCs [87]. The study on patients with breast cancer showed interconnection between the CTC count and survival rate [88]. A similar finding was reported for patients suffering from CRC [89]. In patients with solid tumors, metastasis is the primary cause of death, and CTC quite likely acts as a seed for metastases [90]. For the early diagnosis, recurrence and response to therapy, it may be essential to determine tumor cells or cells with epithelial markers in CRC patients’ peripheral blood [91]. Current research shows that CTC counts are associated with overall and progression-free survival in patients with various metastasizing cancers. CTC count is also considered a reliable indicator of CRC treatment response [87,92,93,94].

Nowadays, many innovative methods are available to detect and analyze CTC, including CTC microchips, filtration devices, molecular analytical methods, CellSearch™ system, flow cytometry methods and automated microscopy [93,94].

Finally, researchers and clinicians can use CTC to identify gene mutations and changes in the signaling pathways, and to monitor malignancy development and response to therapy. The immense benefit of using these approaches is to ensure patients’ safety and reduce manipulative efforts involved in standard diagnostic procedures, which require significantly more resources and time. Therefore, further research and clinical trials are needed to clarify relevant questions and to highlight important clinical aspects of these strategies.

## 8. Conclusions

Taking into consideration and summarizing the molecular disturbances specific to CRC (Figure 1), the present review addresses the usefulness of liquid biopsy in diagnosis of the disease, the choice of efficient treatment, the monitoring of the response to treatment, the progression of the disease, and the detection of recurrence at different molecular levels. Studying various biomarkers (Table 1) and evaluating their potential is crucial to achieving the best therapeutic approaches for patients. Likewise, we presented the contribution of liquid biopsy in providing biological samples (ctDNA, exosomes, CTC) containing potent biomarkers (genes, RNA, proteins), that can give an accurate description and profile of CRC, as well as being used to diagnose and predict disease progression and outcome. We believe that with such a comprehensive approach, one should be able to identify biomarkers useful for CRC diagnosis and predict recurrence and potential for metastasis, monitor the response to treatment and predict outcomes, at least suggestively. A combination of different molecular markers will likely be necessary to make CRC treatment approaches more specific.

## Figures and Tables

**Figure 1 ijms-22-04327-f001:**
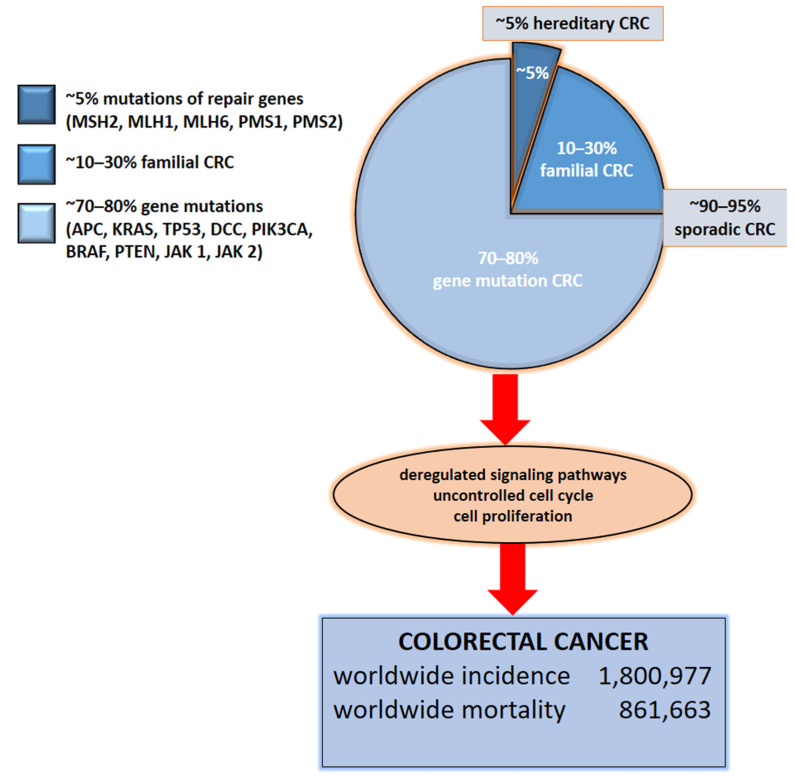
Molecular basis of CRC. Colorectal cancer is based on gene mutations, familial or hereditary CRC. The indication of total numbers refers to Global Cancer Statistics 2018 [3]. For worldwide incidence and mortality, colorectal cancer cases from 185 countries in 2018 were totaled.

**Figure 2 ijms-22-04327-f002:**
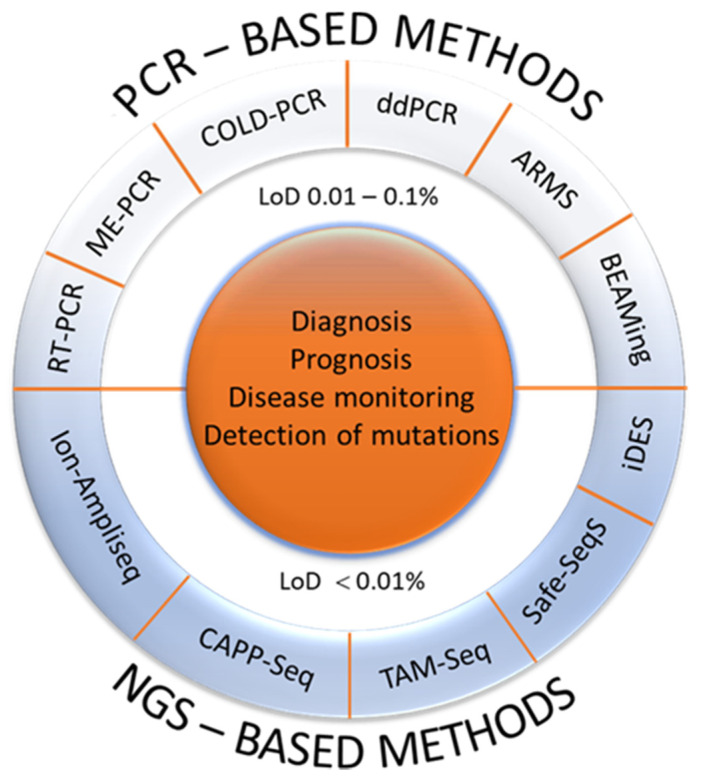
Current methods for identification and analysis of ctDNA. Based on the technologies which are used for ctDNA analysis, PCR- and NGS-based methods emerged. In general, PCR-based methods are cost-effective, rapid and no specific informatic skills are needed but the main disadvantage is that they can detect a limited number of known mutations. On the other hand, NGS is based on the analysis of several millions of short DNA sequences in parallel, followed by either sequence alignments to a reference genome or a de novo sequence assembly. Therefore, the NGS-based methods are expensive and time-consuming, but can detect a large number of mutations. Furthermore, according to the methodological approaches and analytical sensitivity, there are also two strategies for ctDNA analysis: (i) targeted methods with high resolution such as ARMS, ddPCR, BEAMing which in most cases determine only a single or a few mutations with a high analytical sensitivity and (ii) more comprehensive or untargeted, genome-wide approaches, which require a certain amount of tumor DNA in the circulation, typically 5–10%, in order to achieve informative results. To date, there is no consensus regarding methods that could eventually find a practical application since the clinical and practical application depends on individual situation and the goal of the ctDNA analysis [65,66]. (PCR, polymerase chain reaction, RT-PCR, real-time PCR; ME-PCR, mutant-enriched PCR; COLD-PCR, co-amplification at lower denaturation temperature PCR assays; ddPCR, droplet-based digital PCR; ARMS, amplification refractory mutation system; BEAMing, beads-emulsion-amplification-and-magnetics; NGS, next generation sequencing; CAPP-Seq, cancer personalized profiling by deep sequencing; TAM-Seq, tagged-amplicon deep sequencing; Safe-SeqS, Safe-Sequencing System; iDES, integrated digital error suppression; LoD, limit of detection).

**Table 1 ijms-22-04327-t001:** Biomarker usage related to CRC.

Biomarker	Signification	Structure	Experience/Implication	Reference
CEA	carcinoembryonic antigen	glycoprotein	Validated blood biomarker in clinical practice. Not recommended as sole CRC screening test. Preoperative CEA > 5 mg/mL may correlate with poorer CRC prognosis. Used as postoperative serum testing and monitoring during active CRC treatment every 3 months.Diagnostic sensitivity 54.5%; specificity 98.4%.	Locker et al.,2006 [44] Wu et al., 2020 [55]
CA 19-9	carbohydrate antigen	glycoprotein	Validated blood biomarker in clinical practice. Not recommended as sole screening or monitoring CRC marker. Used as supplementary progress monitoring during pancreatic cancer treatment every 1–3 months. Individual values for each patient. Diagnostic sensitivity 64.4%; specificity 96.8%.	Locker et al., 2006 [44] Wu et al., 2020 [55]
CTC	circulating tumor cells	tumor cells	Epithelial marker in the peripheral blood via automatic detection system. Detected in different cancer types. 1-10 CTCs per ml blood were found in patients with metastases but rarely in healthy people. Poor prognosis for CRC patients with ≥ 5 CTC per 7.5 ml blood. Diagnostic sensitivity 62.7%; specificity 82.0%. Multivariate analysis of the disease-free survival data of examined patient group showed that a CTC count ≥5 was an independent prognostic factor of distant metastasis (Hazard ratio = 7.5, 95% CI: 1.6 to 34.7, *p* = 0.01).	Dominguez-Vigil et al., 2018 [49] Tsai et al., 2016 [56]
ctDNA	circulating tumor DNA	small DNA fragments released by tumor cells	Tumor mutation search in the peripheral blood, plasma and serum. Patients with 100 g tumor burden released 3.3% of ctDNA into circulation. In CRC, ctDNA is more sensitive than CEA. KRAS mutations were detected with 87.2% sensitivity and a 99.2% specificity.	Osumi et al., 2020 [57] Said et al., 2020 [58] Dominguez-Vigil et al., 2018 [49]
exosomes	nanovesicles	vesicular structures released by different cell types, including tumors	Tumor miRNA molecules in biological fluid like blood and urine. Associated with several types of CRC. Each tumor is characterized by specific protein profile. Positive correlation from miRNA exosomes and proteins with the stage of tumor progression.	Dominguez-Vigil et al., 2018 [49] Wang et al., 2016 [59]

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
