# Peer review of "Profiling Colorectal Cancer in the Landscape Personalized Testing—Advantages of Liquid Biopsy"

_ijms, 2021, doi:10.3390/ijms22094327_

Round 1

Reviewer 1 Report

The authors discussed about recent molecular approaches in diagnosis of CRC. I think some major points should be improved. Followings are my specific comments.

  1. The etiology and genetic mutation of cancer driver genes, such as KRAS and APC, in colorectal cancer has already been reviewed in numerous papers. These paragraphs should be simplified because the main topics of this manuscript is “Advantages of Liquid Biopsy”. Likewise, the authors must discuss the tumor specific liquid component, such as ctDNA, RNA, exosome and CTC, in more detail.
  2. KRAS inhibitor which targets the KRAS G12C mutation has been reported in previous study (Canon J, Nature 2019). The authors should mention about this.
  3. To discuss the utility of extracellular DNA, RNA and small vesicles in CRC treatment, the list of each molecules which corelate with the tumor progression must be needed.
  4. Discuss about the advantages and/or disadvantages of novel types of CRC biomarker candidates (circulating DNA, exosomal component and CTC) and current clinical tumor biomarker, CEA and CA 19-9, in tissue specificity, accuracy and efficacy of detection of CRC.

Reviewer 2 Report

This review article describes the biomarkers for colorectal cancer, apparently focusing on personalized testing using liquid biopsy based on the title. Overall it provides useful information on the current status of CRC diagnosis with a comprehensive review of the genes and pathways involved in CRC. Although the title includes “personalized” and “advantages of liquid biopsy,” the text does not seem to highlight the personalized feature or the particular advantages of liquid biopsy.

Major

  1. Survival: most patients die in an advanced phase, meaning that they know they are CRC patients. The question is the value of detecting CRC using a more sophisticated set of biomarkers for improving the survival of patients. Nine drugs are listed in the introduction for treating CRC. The authors should show the scenario to improve the survival rate by the profiling of colorectal cancer. For instance, drug X should be used for patients that show a high level of a biomarker Y.
  2. Personalized testing: To achieve personalized testing, it seems important to establish a relationship between the biomarkers for individual patients and the treatment. The treatment should depend on the biomarkers. It is recommended to describe this relationship.
  3. Liquid biopsy: There are several different forms of liquid biopsy besides blood. Most notably, urine and saliva can be important body fluids. The authors may need to state that the review is for blood if no other samples are discussed.
  4. Genes and pathways: The descriptions of genes and pathways are detailed and informative. However, the current descriptions alone are not sufficient for the general readers to have a summary view. It is recommended to include a table that summarizes genes and pathways.
  5. Colorectal cancer: In an entire description, it is difficult to distinguish which part is CRC specific and others are solid tumor in general. The authors should describe and stress CRC-specific pathways and genes in comparison with other solid tumors, for instance, breast cancer and prostate cancer.

Minor

  1. Figure 1: The figure is not informative. Figure legend does not explain the context. Use of % without showing the total number is not a good practice. The % range of 1 to 5 seems too large, although it may depend on the total number.
  2. Figure 2: Figure 2 does not provide sufficient information. The listing of the procedures without any linkage, accuracy, reliability, cost, etc. does not seem to be informative. No detailed information for these techniques is provided.
  3. Table 1: The information in Table 1 is not sufficient. The question is specificity/selectivity to CRC, and the threshold value for positive/negative, reliability, etc.
  4. CTC: CTC is usually detected using an antibody to EpCAM, which is a marker for epithelial cells. When tumor cells experience EMT, then mesenchymal-type of CTC may not be detected by the regular CTC detection. The authors should clarify the method of CTC identification and the value of this technique for predicting risk of metastasis.

Round 2

Reviewer 1 Report

The authors adequately, if not completely, addressed my concerns. The manuscript is suited for publication in International Journal of Molecular Sciences.

Reviewer 2 Report

The authors resonded satisfactorily to the comments, given to the original manuscript. Thank you.